materials science/inorganic chemistry/nanotechnology

SiO₂, core-shell-shell structure, rare earth phosphate, luminescence, nanoparticles

**Author for correspondence:**
Jinrong Bao
e-mail: jinrongbao@imu.edu.cn

†Present address: School of Chemistry and Chemical Engineering, Inner Mongolia University, Hohhot 010021, China.

This article has been edited by the Royal Society of Chemistry, including the commissioning, peer review process and editorial aspects up to the point of acceptance.

# High-efficient fabrication of core-shell-shell structured SiO₂@GdPO₄:Tb@SiO₂ nanoparticles with improved luminescence

He Bai[1], Yunjiang Yang[1], Jinrong Bao[1,†], Anping Wu[1], Yan Qiao[1], Xueyuan Guo[1], Mingyuan Wang[1], Wenxian Li[1], Ying Liu[1] and Xiaowei Zhu[2]

[1]Inner Mongolia Key Laboratory of Chemistry and Physics of Rare Earth Materials, College of Chemistry and Chemical Engineering, and [2]College of Pharmacology, Inner Mongolia Medical University, Hohhot 010110, People's Republic of China

JB, 0000-0002-2480-7330; WL, 0000-0002-0938-6910

SiO₂@GdPO₄:Tb@SiO₂ nanoparticles with core-shell-shell structure were successfully synthesized by a cheap silane coupling agent grafting method at room temperature. This method not only homogeneously coated rare-earth phosphate nanoparticles on the surface of silica spheres but also saved the use of rare-earth resources. The obtained nanoparticles consisted of SiO₂ core with a diameter of approximately 210 nm, GdPO₄:Tb intermediate shell with thickness of approximately 7 nm, and SiO₂ outer shell with thickness of approximately 20 nm. This unique core-shell-shell structured nanoparticles exhibited strong luminescence properties compared with GdPO₄:Tb nanoparticles. The core-shell-shell structured nanoparticles can effectively quench the intrinsic fluorescence of bovine serum albumin through a static quenching mode. The as-synthesized nanoparticles show great potential in biological cell imaging and cancer treatment.

## 1. Introduction

Because of the unique 4f shell of the ions, lanthanide compounds often show good electronic, optical and magnetic characteristics. The lanthanide compounds have attracted considerable interest with major applications in optics, plasma display, drug delivery, magnets and biological labelling [1–4]. A particularly intriguing use of lanthanide phosphate (LnPO₄) is in optoelectronic devices and biological fluorescence labelling, which may be based on its

excellent luminescence, low toxicity, long decay time, and chemical stability [5–7]. Among the rare-earth phosphates, gadolinium phosphate ($GdPO_4$) is an important host matrix for luminescent lanthanide ion-doped nanophosphors [8]. $GdPO_4$ matrix nanoparticles have been proved to be a potential multi-functional nano-platform for magnetic resonance imaging (MR) and effective optical imaging materials [9]. In recent years, gadolinium phosphate nanoparticles have been synthesized by various methods. For example, Rodriguez-Liviano *et al.* have prepared $GdPO_4$:$Eu^{3+}$ nanoparticles via microwave-assisted heating method, which showed potential applications in biolabels [10]. $GdPO_4$:$Eu^{3+}$ nanoparticles also have been synthesized by co-precipitation, and the nanoparticles can emit intense orange-red fluorescence [11]. $GdPO_4$:$Eu^{3+}$/$Tb^{3+}$ was synthesized via typical hydrothermal method; the nanoparticles can obtain a bright colour-tunable photoluminescence from red, orange, yellow to green region when the $GdPO_4$ nanoparticles are co-doped with $Eu^{3+}$ and $Tb^{3+}$ ions [12]. However, most conventional synthesis methods required high temperature, pressure and expensive precursors.

In particular, the lanthanide compounds nano phosphors have poor aqueous solubility and dispersion, and then their biocompatible and biological applications might be limited [13]. In order to enhance the solubility and dispersion in aqueous solution of some namomaterials, the surface of the nanomaterials were modified by polyethylene glycol derivative, poly (acrylic acid) and silica. For example, PEG-$NaYF_4$: Yb/Tm was synthesized by a layer-by-layer strategy, which has great potential in bio-imaging and photodynamic therapy [14]. Hexagonal phase $NaYF_4$:Yb,Er@PAA can be synthesized by the modification method, which has great potential in bio-probes [15]. Xu *et al.* prepared self-assembled Ni/Co phosphide composite N-doped carbon spheres via a hydrothermal process, which had abundant exposed active sites for the hydrogen evolution reaction [16]. The obtained Artemia cyst shell (ACS)–$TiO_2$–$MoS_2$ ternary porous structure has a good reduction effect on 4-NP and 2-NA, which is obviously higher than the reduction effect of ACS–$TiO_2$ and $MoS_2$ under the same conditions [17]. Therefore, the synthesis of $SiO_2$-lanthanide phosphate/oxide nanomaterials with core-shell structure has attracted considerable attention due to the decrease in use of rare earth and its non-toxicity. Core-shell structured $SiO_2$@$Y_2O_3$:$Eu^{3+}$ nanopowder was found to be appropriate as a fluorescent marker for latent fingerprint recognition, security ink and solid-state lighting applications [18]. Xu *et al*. have synthesized a luminescent and mesoporous core-shell structured $Gd_2O_3$:$Eu^{3+}$@$SiO_2$ nanocomposite and make it as a drug carrier [19]. In our previous studies, we have synthesized $SiO_2$@$EuPO_4$ by co-precipitation using triethyl phosphate [20]. It was found that the core-shell nanostructure can significantly improve the emission strength of the material. However, because the hydrolysis rate of tributyl phosphate was not easy to control, the coating uniformity was poor. By bridging ligand organosilane $HOOCC_6H_4N(CONH(CH_2)_3Si(OCH_2CH_3)_3)_2$ (MABA-Si) connected with $SiO_2$ submicrospheres and rare earth ion, it makes $CePO_4$:Tb nanoparticles coated uniformly on the surface of $SiO_2$ submicrospheres. We also synthesized core-shell-shell structured $SiO_2$@$CePO_4$:Tb@$SiO_2$ [21]. In addition, silica shell can greatly improve the stability of $SiO_2$@$GdPO_4$: Tb@$SiO_2$ nanoparticles through protecting the core materials from dissolution or hydrolysis. However, the −$Si(OCH_2CH_3)_3$ group of MABA-Si ligand was easy to hydrolyse in the air, so it is difficult to connect with $SiO_2$ submicrospheres. When $SiO_2$ was used as core and shell of the core-shell-shell particles, it might not only decrease the consumption of rare earth but also give more functions to nanomaterials [21–23].

In this paper, we report a room-temperature silane coupling agent grafting method to simultaneously graft 3-(aminopropyl) triethoxysilane (APTES) on the surface of the silicon spheres and bond with carboxyl of maleic anhydride (MAH). By means of this way, the reaction of silane coupling agent APTES connected with $SiO_2$ spheres and rare-earth phosphate is easy to carry out. Furthermore, nano rare-earth phosphate can be homogeneously coated on the surface of silica spheres. The obtained $SiO_2$@$GdPO_4$:Tb@$SiO_2$ nanoparticles show a core-shell-shell structure with uniform size and coating layer. The $SiO_2$ can be functioned as fixed centre core and protected layer shell, respectively. These unique structures endow $SiO_2$@$GdPO_4$:Tb@$SiO_2$ nanoparticles good luminescence properties. Moreover, the interaction between the core-shell-shell structured nanoparticles and BSA in the simulated physiological conditions was studied. The core-shell-shell structured $SiO_2$@$GdPO_4$:Tb@$SiO_2$ nanoparticles makes nanoparticles highly biocompatible and non-toxic, which would expand their potential applications in the field of biomedicine.

# 2. Material and methods

## 2.1. Material and reagents

All chemicals were analytical, unpurified and used as received. Ammonia, $Tb_4O_7$ (99.99%), $Gd(NO_3)_3$·$6H_2O$, $(NH_4)_2HPO_4$ and cetyltrimethyl ammonium bromide (CTAB) were all purchased by

Shanghai McLean Biochemical Technology Corporation Limited. 3-(aminopropyl) triethoxysilane (APTES), MAH and tetraethoxysilane (TEOS) were achieved from Aladdin (Shanghai, China). Bovine serum albumin (BSA, biochemical reagent, average molecular weight of $66\,000\,\mathrm{g\,mol^{-1}}$) was supplied by Beijing bailingwei Technology Corporation Limited (Beijing, China). The terbium nitrate powder prepared from $Tb_4O_7$ was dissolved in 10% nitric acid, then evaporated and dried in vacuum.

## 2.2. Synthesis of $SiO_2@GdPO_4$:Tb@$SiO_2$ nanoparticles

The core-shell structured $SiO_2@GdPO_4$:Tb was prepared by the following steps. The $SiO_2$ spheres were synthesized by the Stöber method [24], in which 0.2 g were dispersed in anhydrous ethanol via ultrasonication. Then 0.5 ml APTES was put into above ethanol suspension under stirring for 12 h. After centrifugation, the above as-prepared product (labelled as $SiO_2@NH_2$) was dispersed in ethanol, and dropped in 1.5 mmol MAH ethanol solution stirring for 6.0 h. The obtained solution (labelled as $SiO_2@MAH$-Si) was centrifuged. Then, it was dispersed in 10 ml anhydrous ethanol followed by adding of $0.098\,\mathrm{mol\,l^{-1}}$ $Ln(NO_3)_3$ ($Gd^{3+}$ 95%, $Tb^{3+}$ 5%) ethanol solution, which was further stirred for 4 h. Finally, 0.0216 g $(NH_4)_2HPO_4$ was added, and continuously reacted for 2 h. The $SiO_2@GdPO_4$:Tb was obtained by further centrifugation and washing with ethanol three times.

For the synthesis of the $SiO_2@GdPO_4$:Tb@$SiO_2$ nanoparticles, the above as-prepared $SiO_2@GdPO_4$:Tb was dispersed in 20 ml 50% ethanol solution via ultrasonication. Then, 0.15 g of cetyltrimethyl ammonium bromide and 0.3 ml of tetraethoxysilane (TEOS) was added to above suspension under stirring for 6 h. After centrifugation, the obtained white solid were further washed with ethanol three times. The white solid was dried at 80°C for 6 h, which was then treated at 600°C for 2 h under nitrogen atmosphere.

## 2.3. Interaction between BSA and $SiO_2@GdPO_4$:Tb@$SiO_2$ nanoparticles

The whole BSA binding experiment was performed in Tris–HCl buffer solution with pH = 7.4. The solutions of BSA and the core-shell-shell structured $SiO_2@GdPO_4$:Tb@$SiO_2$ nanoparticles were prepared by dissolving them in the Tris–HCl buffer solution to obtain the desired concentrations. In the fluorescence quenching experiment of BSA, the quenching of BSA was achieved by keeping BSA as a fixed concentration and adding core-shell nanoparticles with different concentrations ($a = 0.000$, $b = 1.85 \times 10^{-5}$, $c = 3.70 \times 10^{-5}$, $d = 5.55 \times 10^{-5}$, $e = 7.42 \times 10^{-5}$, $f = 9.25 \times 10^{-5}$, $g = 1.11 \times 10^{-4}$, $h = 1.29 \times 10^{-4}$, $i = 1.48 \times 10^{-4}$ and $j = 1.66 \times 10^{-4}$ $\mathrm{mol\,l^{-1}}$). Fluorescence measurements were made at 293 K, 303 K and 313 K. The fluorescence spectra of BSA were tested at an excitation wavelength at 280 nm and an emission wavelength at 335 nm after addition of the core-shell-shell nanoparticles.

## 2.4. Characterization

The morphology of the products was characterized by transmission electron microscopy (TEM; FEI Tecnai F20, USA) and scanning electronic microscopy (SEM; Hitachi S-4800, Japan). The crystal structure is investigated by X-ray powder diffraction (XRD; RIGAKU, Japan) using Cu $K_\alpha$ radiation. Infrared spectrum of the solid powders was determined in the range of 400–4000 $\mathrm{cm^{-1}}$ (FT-IR; Bruker, Germany). The luminescence spectra of powders was examined on a fluorescence photometer (FL; Edinburgh S980, UK).

# 3. Results and discussion

XRD analysis investigated the phase purity and crystal structure of the as-prepared products. Figure 1 shows the XRD patterns of $SiO_2$ and $SiO_2@GdPO_4$:Tb@$SiO_2$ nanoparticles. It can be seen that two diffraction peaks at $2\theta = 8°$ and 22° from amorphous $SiO_2$ were detected on both samples. Several new weak diffraction peaks appeared in $SiO_2@GdPO_4$:Tb@$SiO_2$, which were matched with monoclinic phase of $GdPO_4$ (JCPDS No. 32–386). The microstructure and size of the as-obtained samples were examined from TEM images as shown in figure 2. TEM image of $SiO_2$ (figure 2a) and the particle size distribution indicated that $SiO_2$ spheres have a regular morphology and excellent monodispersity with diameters about 210 nm. When $SiO_2$ were coated with $GdPO_4$:Tb, the surface of the obtained $SiO_2@GdPO_4$:Tb spheres becomes rough and the diameter of $SiO_2@GdPO_4$:Tb is about 225 nm. To make the nanoparticles more functional, the surface of the $SiO_2@GdPO_4$:Tb spheres were modified by

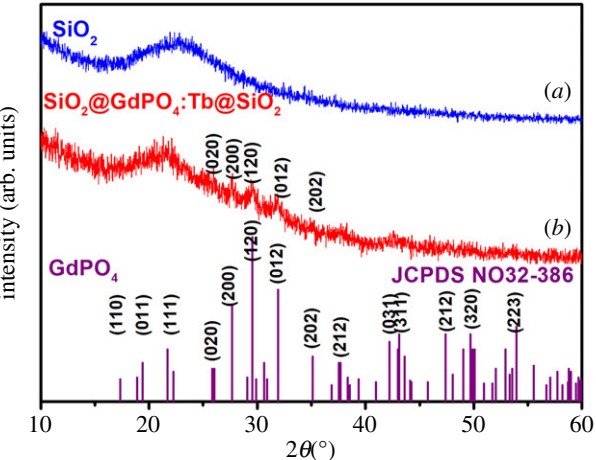

**Figure 1.** XRD patterns of (*a*) SiO$_2$, (*b*) SiO$_2$@GdPO$_4$:Tb@SiO$_2$.

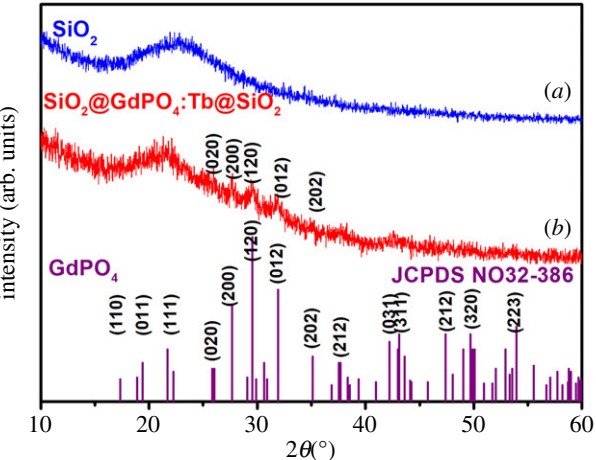

**Figure 2.** TEM images of the products (*a*) SiO$_2$, (*b*) SiO$_2$@GdPO$_4$:Tb, (*c–f*) SiO$_2$@GdPO$_4$:Tb@SiO$_2$, and corresponding size distribution images.

SiO$_2$ as shown in figure 2*c–f* at different magnification. It can be shown that the product SiO$_2$@GdPO$_4$: Tb@SiO$_2$ have obvious core-shell-shell structures and smooth surfaces. The corresponding particle size distribution indicated that the core-shell-shell structures have diameters of about 265 nm. The thickness of the intermediate shell GdPO$_4$:Tb was approximately 7 nm, and the diameter of the SiO$_2$ core and outer shell was approximately 210 and approximately 20 nm, respectively. In addition, we

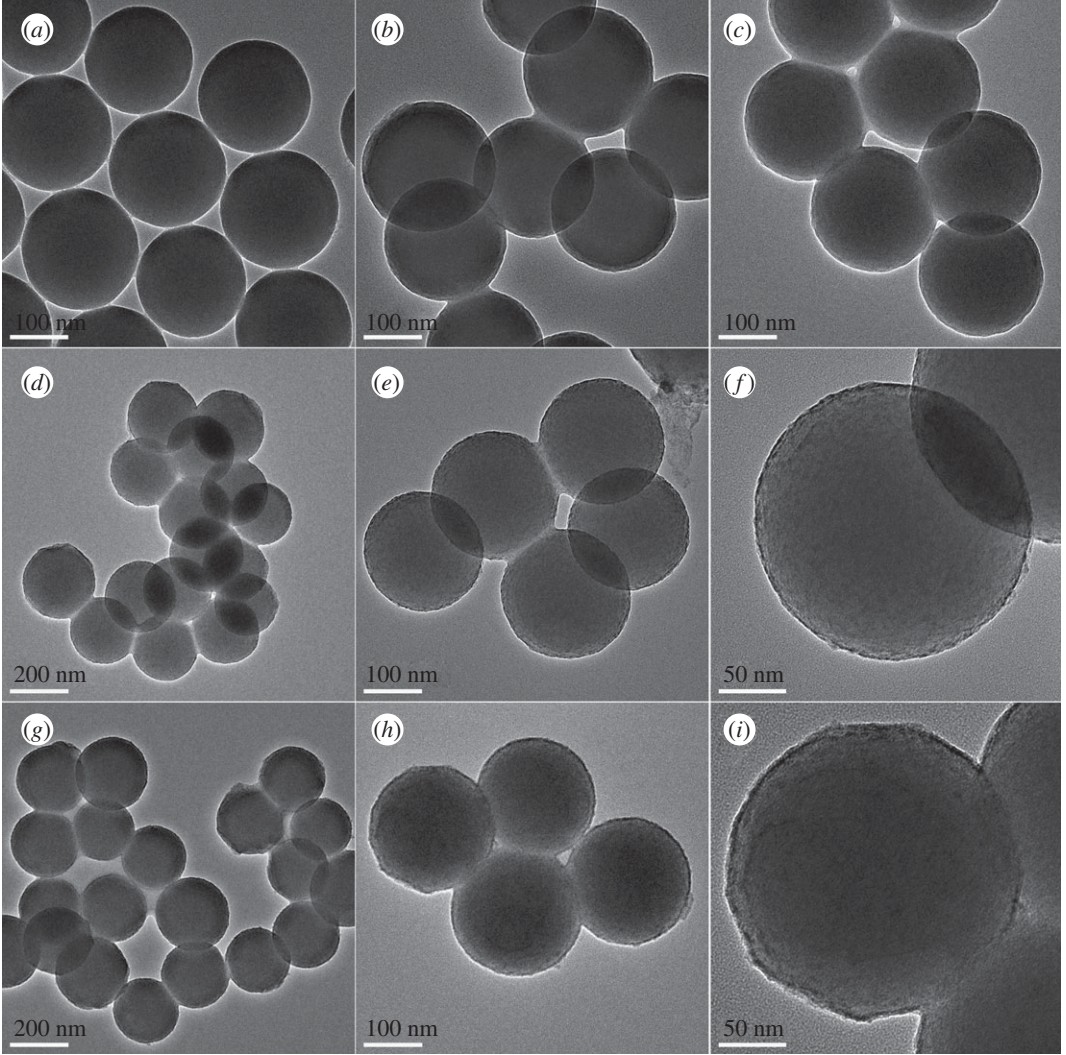

**Figure 3.** TEM images of the products synthesized in different stages: (*a*) SiO₂, (*b*) SiO₂@NH₂, (*c*) SiO₂@MAH-Si, (*d*–*f*) SiO₂@MAH-Si-Gd:Tb, (*g*–*i*) SiO₂@GdPO₄:Tb.

can clearly see that intermediate shell GdPO₄:Tb was uniformly grown on the surface of the SiO₂ core. The corresponding FESEM images of the as-synthesized products are shown in electronic supplementary material, figure S1. It can be seen that SiO₂ spherical particles with an average size of 210 nm were non-aggregated and uniformly distributed (electronic supplementary material, figure S1a and b). The diameter of SiO₂@GdPO₄:Tb increased to 225 nm after GdPO₄:Tb coating, and the surface became rougher (electronic supplementary material, figure S1c and d). Furthermore, SiO₂@GdPO₄: Tb@SiO₂ still maintained a good spherical shape, while the particle size was increased to 265 nm (electronic supplementary material, figure S1e and f). Electronic supplementary material, figure S1e,f shows SiO₂@GdPO₄:Tb@SiO₂ still maintained a good spherical shape with a size of about 265 nm. Meanwhile, SiO₂@GdPO₄:Tb@SiO₂ nanoparticles also had a high BET surface area of 62 m² g⁻¹ (electronic supplementary material, figure S2).

We further characterized the products at various synthesis stages by TEM and IR. Firstly, figure 3*a* demonstrates that the surface of SiO₂ spheres obtained from the hydrolysis of TEOS was very smooth, and the diameter of SiO₂ sphere was about 210 nm. In the corresponding IR spectra, the vibration of Si-OH of SiO₂ was found at 952 cm⁻¹ (electronic supplementary material, figure S3a), which would provide active bonds for grafting. Secondly, after APTES was grafted on the surface of SiO₂ spheres through Si-O-Si bond that of −NH₂ group appeared at 1640 cm⁻¹ (electronic supplementary material, figure S3b), figure 3*b* demonstrates that the diameters of the SiO₂@NH₂ nanoparticles further increased approximately 2 nm. Thirdly, the SiO₂@MAH-Si nanoparticles were obtained by APTES bonded with MAH. Figure 3*c* demonstrates that there was no obvious change in thickness after

**Figure 4.** Schematic illustration showing the formation mechanism of core-shell-shell structured $SiO_2@GdPO_4:Tb@SiO_2$.

APTES bonded with MAH. However, three new stretching vibration peaks including −COOH group at 1728 cm$^{-1}$ and −CONH- group at 1632 and 1596 cm$^{-1}$ appeared, which suggested that there was a bond connection between MAH and APTES (electronic supplementary material, figure S3c). TEM images of figure 3d–f show that plenty of small particles grown on the surface of $SiO_2$@MAH-Si nanoparticles after rare-earth ions were coordinated with −COOH of MAH. Simultaneously, the stretching vibration peak of −COOH group was shifted to 1720 cm$^{-1}$ (electronic supplementary material, figure S3d). In the next step, the $GdPO_4$:Tb nanoparticles were formed on the surface of $SiO_2$ spheres by the substitution reaction of $PO_4^{3-}$. TEM images of figure 3g–i show that the surface of $SiO_2@GdPO_4$:Tb nanoparticles became rough and the rough layer thickness was approximately 7 nm. Therefore, it is reasonable to conclude that $GdPO_4$:Tb layer with thickness of approximately 7 nm have been uniformly coated on the $SiO_2$ core through substitution reaction. Finally, $SiO_2$ outer shell was covered on the surface of $SiO_2@GdPO_4$:Tb nanoparticles in the presence of CTAB, through the hydrolysis process of TEOS. CTAB formed a molecular layer on the surface of the silicon core in the reaction system, which would guarantee uniform hydrolysis and growth for TEOS. After calcination, the core-shell-shell structured $SiO_2@GdPO_4:Tb@SiO_2$ nanoparticles were obtained (figures 1 and 2). Energy-dispersive X-ray spectroscopy (EDX) of core-shell-shell structured $SiO_2@GdPO_4:Tb@SiO_2$ was conducted (electronic supplementary material, figure S4); it can be clearly seen that the weight percentages of Si, P, Gd and Tb are 36.73%, 0.74%, 5.79% and 0.29%, respectively. The schematic of core-shell-shell structured $SiO_2@GdPO_4:Tb@SiO_2$ formation process is illustrated in figure 4. Furthermore, functionalized $SiO_2@GdPO_4:Tb@SiO_2$ nanoparticles can be reused after calcination. In other words, the adsorbed proteins and biomolecules can be removed from the surface of nanoparticles after heat treatment of the functionalized $SiO_2@GdPO_4:Tb@SiO_2$ nanoparticles.

The luminescence property of the core-shell-shell structured $SiO_2@GdPO_4:Tb@SiO_2$ and $GdPO_4$:Tb nanoparticles with prepared hydrothermal method was investigated at room temperature. Excitation spectra showed that the strongest excitation peak of $GdPO_4$:Tb nanoparticles appeared at 273 nm, while the core-shell-shell structured $SiO_2@GdPO_4:Tb@SiO_2$ also appeared at 273 nm (figure 5a). When these products were excited at strongest excitation wavelength, the emission peaks centred at 488, 543, 584 and 620 nm, which corresponded to the $^5D_4 \rightarrow {}^7F_6$, $^5D_4 \rightarrow {}^7F_5$, $^5D_4 \rightarrow {}^7F_4$ and $^5D_4 \rightarrow {}^7F_3$ transitions for the $Tb^{3+}$ ion [25], respectively (figure 5b). The emission intensity of $SiO_2@GdPO_4:Tb@SiO_2$ was stronger than that of $GdPO_4$:Tb nanoparticles, which is consistent with the measurement results of the quantum yield. The absolute quantum yields of $SiO_2@GdPO_4:Tb@SiO_2$ and $GdPO_4$:Tb were 28.28% and 2.73%, respectively. Meanwhile, the photoluminescence lifetime of the products was also measured. The photoluminescence lifetimes were calculated through the double exponential mode $(\tau) = (A_1\tau_1^2 + A_2\tau_2^2)/(A_1\tau_1 + A_2\tau_2)$ and $I(t) = I_0 + A_1 \exp(-t_1/\tau_1) + A_2 \exp(-t_2/\tau_2)$. Where $I(t)$ is the photoluminescence intensity, $\tau_1$ and $\tau_2$ stand for the slow and fast terms of the luminescent lifetime, respectively. $A_1$ and $A_2$ are the corresponding pre-exponential factors. The average lifetime ($\tau$) of the $SiO_2@GdPO_4:Tb@SiO_2$ and $GdPO_4$:Tb calculated from their fluorescence decay curves shown in electronic supplementary material, figure S5 were 1.38 and 2.18 ms, respectively. The rare-earth

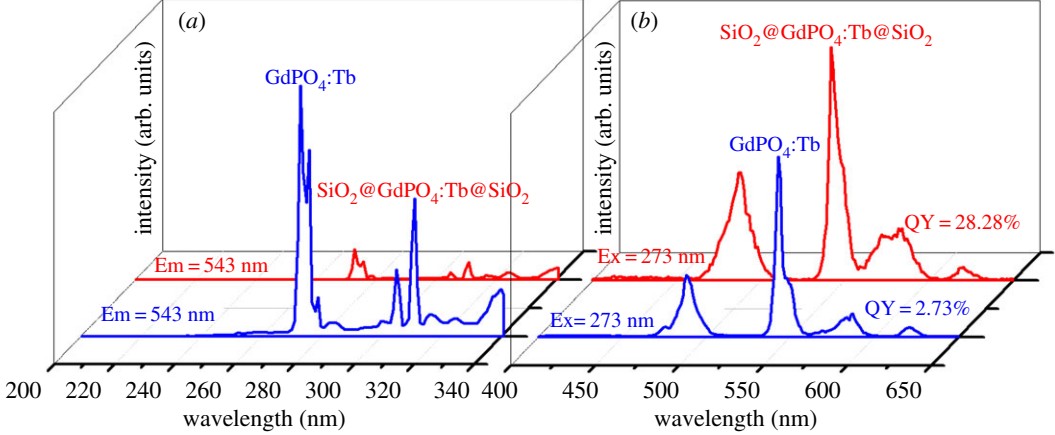

**Figure 5.** Excitation spectra (*a*) and emission spectra (*b*) of blue line (GdPO$_4$:Tb) and red line (SiO$_2$@GdPO$_4$:Tb@SiO$_2$).

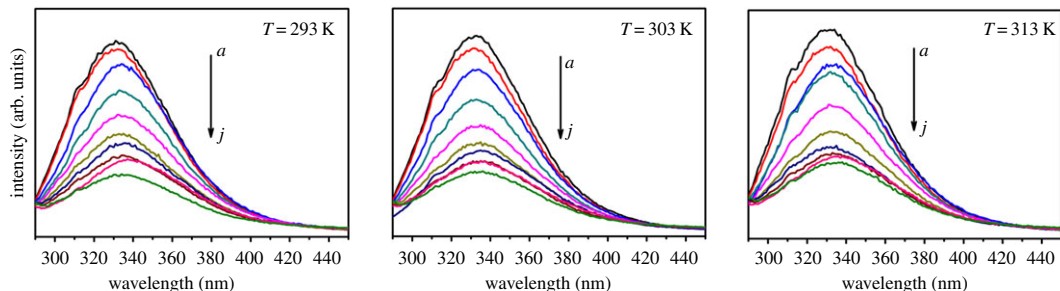

**Figure 6.** Fluorescence quenching spectra of BSA with various concentrations (*a* = 0.000, *b* = 1.85 × 10$^{-5}$, *c* = 3.70 × 10$^{-5}$, *d* = 5.55 × 10$^{-5}$, *e* = 7.42 × 10$^{-5}$, *f* = 9.25 × 10$^{-5}$, *g* = 1.11 × 10$^{-4}$, *h* = 1.29 × 10$^{-4}$, *i* = 1.48 × 10$^{-4}$, and *j* = 1.66 × 10$^{-4}$ mol l$^{-1}$) of SiO$_2$@GdPO$_4$:Tb@SiO$_2$ nanoparticles at different temperature ($\lambda_{ex}$ = 280 nm). *c*(BSA) = 1.0 × 10$^{-7}$ mol l$^{-1}$.

phosphate was protected from the perturbation of the surrounding environment, which contributed by SiO$_2$ shell. We have also studied the interaction between the as-prepared SiO$_2$@GdPO$_4$:Tb@SiO$_2$ nanoparticles and the BSA. The fluorescence spectra of BSA with increase of the SiO$_2$@GdPO$_4$: Tb@SiO$_2$ nanoparticles concentration at different temperature were measured as shown figure 6. It can be shown that the emission intensity of BSA decreases along with the increase of the nanoparticles concentration, which indicates that the intrinsic fluorescence of BSA can be quenched by adding SiO$_2$@GdPO$_4$:Tb@SiO$_2$ nanoparticles into BSA solution. Usually, the main fluorescence quenching mechanism of BSA is dynamics quenching or static quenching. The types of fluorescence quenching mechanisms can be distinguished by different dependence on temperature [26,27].

The fluorescence quenching intensities at 335 nm for the BSA plus SiO$_2$@GdPO$_4$:Tb@SiO$_2$ nanoparticles system at 293, 303 and 313 K were fitted by the below Stern–Volmer equation (3.1) [27,28]

$$\frac{F_0}{F} = 1 + K_q\tau_0[Q] = 1 + K_{sv}[Q], \tag{3.1}$$

where $F_0$ and $F$ are the emission intensities of BSA and BSA with nanoparticles, respectively; $K_q$ is the maximum scatter collision quenching constant; $\tau_0$ is the lifetime of the BSA, the value is approximately 10$^{-8}$ s; $K_{sv}$ is the Stern–Volmer quenching constant and [Q] is the concentration of SiO$_2$@GdPO$_4$:Tb@SiO$_2$ nanoparticles [29].

For this system, the $K_{sv}$ could be obtained from the Stern–Volmer equation (3.1). The graph of $F_0/F$ against [Q] at 293, 303 and 313 K were plotted (figure 7) and the corresponding data were summarized in table 1 for the quenching of BSA by SiO$_2$@GdPO$_4$:Tb@SiO$_2$ nanoparticles. The calculated values of $K_{sv}$ were 1.0292 × 10$^4$ at 293 K ($R^2$ = 0.975), 1.0148 × 10$^4$ at 303 K ($R^2$ = 0.981) and 0.9189 × 10$^4$ l mol$^{-1}$ at 313 K ($R^2$ = 0.979). The value of $K_{sv}$ was decreased with rising temperature. It can be preliminarily estimated that the fluorescence quenching mechanism of BSA by SiO$_2$@GdPO$_4$:Tb@SiO$_2$ nanoparticles was initiated by the formation of a SiO$_2$@GdPO$_4$:Tb@SiO$_2$-protein complex. The fluorescence quenching mechanism of BSA by SiO$_2$@GdPO$_4$:Tb@SiO$_2$ nanoparticles was static quenching [30]. At

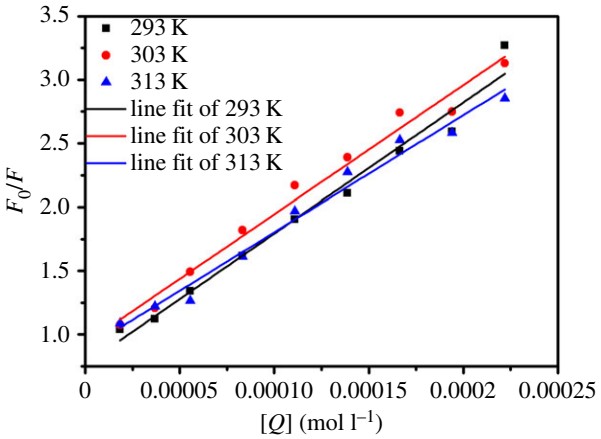

**Figure 7.** Stern–Volmer plots for the quenching of BSA by $SiO_2@GdPO_4{:}Tb@SiO_2$ at different temperature.

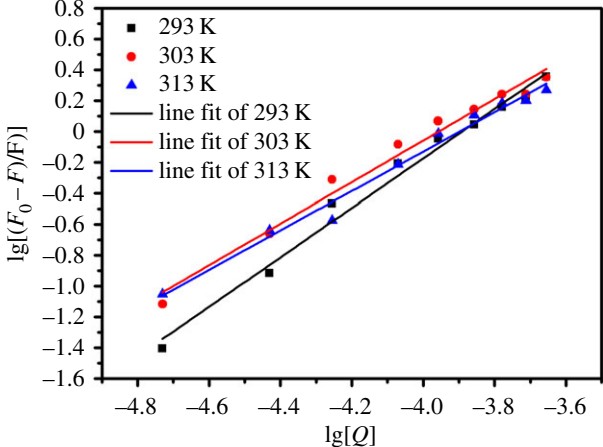

**Figure 8.** Relation curves of $lg[(F_0 - F)/F)]$ and $lg[Q]$ of $SiO_2@GdPO_4{:}Tb@SiO_2$.

**Table 1.** The parameters of Stern–Volmer plots for the fluorescence quenching of BSA by $SiO_2@GdPO_4{:}Tb@SiO_2$ at different temperature.

| $T(K)$ | Stern–Volmer linear equation | $K_{sv}$ ($l\,mol^{-1}$) | $K_q$ ($\times 10^{-8}\,l\,mol^{-1}$) | $R^2$ |
| --- | --- | --- | --- | --- |
| 293 | $F_0/F = 0.7637 + 1.0292[Q]$ | $1.0292 \times 10^4$ | $1.0292 \times 10^4$ | 0.975 |
| 303 | $F_0/F = 0.9294 + 1.0148[Q]$ | $1.0148 \times 10^4$ | $1.0148 \times 10^4$ | 0.981 |
| 313 | $F_0/F = 0.8842 + 0.9189[Q]$ | $0.9189 \times 10^4$ | $0.9189 \times 10^4$ | 0.979 |

the same time, we used the following Mineweaver–Burk curve equation to calculate the binding constants ($K_a$) and binding sites ($n$) for the BSA plus $SiO_2@GdPO_4{:}Tb@SiO_2$ nanoparticles:

$$lg\left[\frac{F_0 - F}{F}\right] = lg\,K_a + n\,lg\,[Q], \tag{3.2}$$

where $F$, $F_0$ are the emission intensities of BSA and BSA with nanoparticles, and $[Q]$ is the concentration of $SiO_2@GdPO_4{:}Tb@SiO_2$ nanoparticles. The relation curves of $lg[(F_0 - F)/F)]$ and $lg[Q]$ at 293, 303 and 313 K for $SiO_2@GdPO_4{:}Tb@SiO_2$ nanoparticles are shown in figure 8. The values of $K_a$ and $n$ at different temperature were measured from the intercept and slope values by the relation curves of $lg[(F_0 - F)/F)]$ and $lg[Q]$ (listed in table 2). According to table 2, $K_a$ = $1.682 \times 10^6$ (293 K), $2.082 \times 10^5$ (303 K) and $9.313 \times 10^4\,l\,mol^{-1}$ (313 K) and $n$ = 1.6002 (293 K), 1.3443 (303 K) and 1.2748 (313 K) for BSA-$SiO_2@GdPO_4{:}Tb@SiO_2$ nanoparticles system, respectively. It could be seen that the binding constants were

**Table 2.** Binding constants ($K_a$) and binding sites ($n$) of $SiO_2@GdPO_4$:Tb@$SiO_2$ nanoparticles with BSA at different temperature.

| $T(K)$ | equation | $K_a$ ($l\ mol^{-1}$) | $n$ | $R^2$ |
|---|---|---|---|---|
| 293 | $lg[(F_0 - F)/F)] = 6.2264 + 1.6002 lg[Q]$ | $1.682 \times 10^6$ | 1.6002 | 0.984 |
| 303 | $lg[(F_0 - F)/F)] = 5.3185 + 1.3443 lg[Q]$ | $2.082 \times 10^5$ | 1.3443 | 0.979 |
| 313 | $lg[(F_0 - F)/F)] = 4.9691 + 1.2748 lg[Q]$ | $9.313 \times 10^4$ | 1.2748 | 0.982 |

decreased with rising the temperature, which indicates that the binding ability of BSA and $SiO_2@GdPO_4$:Tb@$SiO_2$ nanoparticles decreased.

## 4. Conclusion

The core-shell-shell structured $SiO_2@GdPO_4$:Tb@$SiO_2$ nanoparticles with uniform coating layer have been successfully synthesized at room temperature. The possible growth mechanism of $SiO_2@GdPO_4$:Tb@$SiO_2$ nanoparticles was proposed. The $SiO_2@GdPO_4$:Tb@$SiO_2$ nanoparticles have strong green luminescence. Interestingly, the emission intensity and the absolute quantum yield of $GdPO_4$:Tb nanoparticles were improved by the $SiO_2$ shell. The absolute quantum yield of $SiO_2@GdPO_4$:Tb@$SiO_2$ is about 10 times higher than that of $GdPO_4$:Tb nanoparticles. The interaction between the core-shell-shell structured nanoparticles and BSA was investigated through the fluorescence spectroscopy. The quenching mechanism of the fluorescence of BSA by $SiO_2@GdPO_4$:Tb@$SiO_2$ nanoparticles can be attributed to the static quenching.

Data accessibility. Our data are deposited at the Dryad Digital Repository: https://doi.org/10.5061/dryad.ttdz08ktf [31].
Authors' contributions. J.B. designed research; H.B., X.G and M.W. performed the experimental work. H.B. wrote the manuscript. H.B., Y.Y., J.B., A.W., Y.Q., X.G., M.W., W.L., Y.L., X.Z., contributed to the scientific discussion of the results. All authors gave final approval for publication.
Competing interests. The authors declare no competing interests.
Funding. This work was supported by the National Natural Science Foundations of China (21766021) and the Scientific Research Project of Colleges and Universities in Inner Mongolia Autonomous region (NJZZ19002). The project was also funded by the major basic research and open project of the Inner Mongolia Autonomous Region (30500-515330303).
Acknowledgements. All the people who contributed to the study are listed as co-authors.

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
