## [Reviewer comments · Royal Society Open Science]

Review History

RSOS-192235.R0 (Original submission)

Review form: Reviewer 1

Is the manuscript scientifically sound in its present form?

No

Are the interpretations and conclusions justified by the results?

No

Is the language acceptable?

No

Do you have any ethical concerns with this paper?

Yes

Have you any concerns about statistical analyses in this paper?

No

Recommendation?

Reject

Comments to the Author(s)

The paper content is confusing and has many questions. The paper may not be accepted.

In Fig. 1, the XRD patterns of $\text{SiO}_2@\text{GdPO}_4:\text{Tb}@\text{SiO}_2$ is difficult to confirm the successful synthesis of samples because they do not match.

Excitation spectra showed that the strongest excitation peak of $\text{GdPO}_4:\text{Tb}$ nanoparticles appeared at 273 nm, while the core-shell-shell structured $\text{SiO}_2@\text{GdPO}_4:\text{Tb}@\text{SiO}_2$ appeared at 301 nm.

Why?

Excitation spectra should be shown in Fig. 5.

In Fig. S3, the excitation spectrum of $\text{SiO}_2@\text{GdPO}_4:\text{Tb}@\text{SiO}_2$ is questionable. It should be retested.

In Fig. 5, the excitation wavelength of $\text{GdPO}_4:\text{Tb}$ and $\text{SiO}_2@\text{GdPO}_4:\text{Tb}@\text{SiO}_2$ should all be 301 nm for their comparison.

Why is the excitation wavelength 280 nm in Fig. 6?

What are the monitoring and excitation wavelength in Fig. S4?

Review form: Reviewer 2**Is the manuscript scientifically sound in its present form?**

Yes

Are the interpretations and conclusions justified by the results?

No

Is the language acceptable?

Yes

Do you have any ethical concerns with this paper?

No

Have you any concerns about statistical analyses in this paper?

No

Recommendation?

Major revision is needed (please make suggestions in comments)

Comments to the Author(s)

This manuscript reports the preparation of $\text{SiO}_2@\text{GdPO}_4:\text{Tb}@\text{SiO}_2$ core-shell-shell structure. The luminescence properties of this unique core-shell-shell structured were studied. The manuscript can be accepted for publication after major revision. But the following contents should be addressed.

1. The authors are encouraged to give the high-magnification and low-magnification FESEM images of samples.
2. In Fig. 4, the schematic diagram of $\text{SiO}_2@\text{NH}_2$ is wrong.
3. What is the role of CTAB in the formation of SiO_2 shells?
4. Experimental data of nitrogen adsorption-desorption should be provided.
5. Fig. S2 is not clear and needs to be redrawn .
6. The authors should discuss in detail the effect of SiO_2 shell thickness on the luminescence performance of samples, and give the experimental results.

Review form: Reviewer 3

Is the manuscript scientifically sound in its present form?

Yes

Are the interpretations and conclusions justified by the results?

Yes

Is the language acceptable?

Yes

Do you have any ethical concerns with this paper?

No

Have you any concerns about statistical analyses in this paper?

No

Recommendation?

Accept with minor revision (please list in comments)

Comments to the Author(s)

In this manuscript the authors reported the investigation of SiO₂@GdPO₄:Tb@SiO₂ nanoparticles with core-shell-shell structure by a silane coupling agent grafting method. Some interesting results are obtained. The core-shell-shell structured nanoparticles can effectively quench the intrinsic fluorescence of BSA through a static quenching mode. I therefore recommend an acceptance for publishing after next revisions.

1. Pages 2, Summary part, some background sentences can be simplified;
2. Introduction part, if possible, some important and relative reports about self-assembled core-shell-shell nanostructures (COMPOSITES PART B-ENGINEERING 2019,164 : 324-332; Applied Surface Science, 2020, 509: 145383.; Nanomaterials 2020, 10(1): 1.; Nanotechnology, 2020, 31(8): 085603.; Journal of Molecular Liquids, 2020, 298: 112010.) should be added to show clear background;
3. GA image should be added;
4. what about the stability and reuse for this composite materials, pleas add more describe?
5. Some minor Language error should be modified;

Decision letter (RSOS-192235.R0)

30-Jan-2020

Dear Dr Bao:

Title: High-efficient fabrication of core-shell-shell structured SiO₂@GdPO₄:Tb@SiO₂ nanoparticles with improved luminescence
Manuscript ID: RSOS-192235

The editor assigned to your manuscript has now received comments from reviewers. We would like you to revise your paper in accordance with the referee and Subject Editor suggestions which

can be found below (not including confidential reports to the Editor). Please note this decision does not guarantee eventual acceptance.

Please submit your revised paper before 22-Feb-2020. Please note that the revision deadline will expire at 00.00am on this date. If we do not hear from you within this time then it will be assumed that the paper has been withdrawn. In exceptional circumstances, extensions may be possible if agreed with the Editorial Office in advance. We do not allow multiple rounds of revision so we urge you to make every effort to fully address all of the comments at this stage. If deemed necessary by the Editors, your manuscript will be sent back to one or more of the original reviewers for assessment. If the original reviewers are not available we may invite new reviewers.

RSC Associate Editor:
Comments to the Author:
(There are no comments.)

RSC Subject Editor:
Comments to the Author:
(There are no comments.)

Reviewers' Comments to Author:

Reviewer: 1

Comments to the Author(s)

The paper content is confusing and has many questions. The paper may not be accepted.

In Fig. 1, the XRD patterns of SiO₂@GdPO₄:Tb@SiO₂ is difficult to confirm the successful synthesis of samples because they do not match.

Excitation spectra showed that the strongest excitation peak of GdPO₄:Tb nanoparticles appeared at 273 nm, while the core-shell-shell structured SiO₂@GdPO₄:Tb@SiO₂ appeared at 301 nm.

Why?

Excitation spectra should be shown in Fig. 5.

In Fig. S3, the excitation spectrum of SiO₂@GdPO₄:Tb@SiO₂ is questionable. It should be retested.

In Fig. 5, the excitation wavelength of GdPO₄:Tb and SiO₂@GdPO₄:Tb@SiO₂ should all be 301 nm for their comparison.

Why is the excitation wavelength 280 nm in Fig. 6?

What are the monitoring and excitation wavelength in Fig. S4?

Reviewer: 2

Comments to the Author(s)

This manuscript reports the preparation of SiO₂@GdPO₄:Tb@SiO₂ core-shell-shell structure. The luminescence properties of this unique core-shell-shell structured were studied. The manuscript can be accepted for publication after major revision. But the following contents should be addressed.

1. The authors are encouraged to give the high-magnification and low-magnification FESEM images of samples.
2. In Fig. 4, the schematic diagram of SiO₂@NH₂ is wrong.
3. What is the role of CTAB in the formation of SiO₂ shells?
4. Experimental data of nitrogen adsorption-desorption should be provided.
5. Fig. S2 is not clear and needs to be redrawn .
6. The authors should discuss in detail the effect of SiO₂ shell thickness on the luminescence performance of samples, and give the experimental results.

Reviewer: 3

Comments to the Author(s)

In this manuscript the authors reported the investigation of SiO₂@GdPO₄:Tb@SiO₂ nanoparticles with core-shell-shell structure by a silane coupling agent grafting method. Some interesting results are obtained. The core-shell-shell structured nanoparticles can effectively quench the intrinsic fluorescence of BSA through a static quenching mode. I therefore recommend an acceptance for publishing after next revisions.

1. Pages 2, Summary part, some background sentences can be simplified;
2. Introduction part, if possible, some important and relative reports about self-assembled core-shell-shell nanostructures (COMPOSITES PART B-ENGINEERING 2019,164 : 324-332; Applied Surface Science, 2020, 509: 145383.; Nanomaterials 2020, 10(1): 1.; Nanotechnology, 2020, 31(8): 085603.; Journal of Molecular Liquids, 2020, 298: 112010.) should be added to show clear background;
3. GA image should be added;
4. what about the stability and reuse for this composite materials, pleas add more describe?
5. Some minor Language error should be modified

Author's Response to Decision Letter for (RSOS-192235.R0)

See Appendix A.

RSOS-192235.R1 (Revision)

Review form: Reviewer 2

Is the manuscript scientifically sound in its present form?

Yes

Are the interpretations and conclusions justified by the results?

Yes

Is the language acceptable?

Yes

Do you have any ethical concerns with this paper?

No

Have you any concerns about statistical analyses in this paper?

No

Recommendation?

Accept with minor revision (please list in comments)

Comments to the Author(s)

Graphical Abstract needs to be redrawn

Decision letter (RSOS-192235.R1)

26-Mar-2020

Dear Dr Bao:

Title: High-efficient fabrication of core-shell-shell structured $\text{SiO}_2@\text{GdPO}_4:\text{Tb}@\text{SiO}_2$ nanoparticles with improved luminescence
Manuscript ID: RSOS-192235.R1

Thank you for submitting the above manuscript to Royal Society Open Science. On behalf of the Editors and the Royal Society of Chemistry, I am pleased to inform you that your manuscript will be accepted for publication in Royal Society Open Science subject to minor revision in accordance with the referee suggestions. Please find the reviewers' comments at the end of this email.

The reviewers and handling editors have recommended publication, but also suggest some minor revisions to your manuscript. Therefore, I invite you to respond to the comments and revise your manuscript.

Because the schedule for publication is very tight, it is a condition of publication that you submit the revised version of your manuscript before 04-Apr-2020. Please note that the revision deadline will expire at 00.00am on this date. If you do not think you will be able to meet this date please let me know immediately.

To revise your manuscript, log into <https://mc.manuscriptcentral.com/rsos> and enter your Author Centre, where you will find your manuscript title listed under "Manuscripts with

Decisions". Under "Actions," click on "Create a Revision." You will be unable to make your revisions on the originally submitted version of the manuscript. Instead, revise your manuscript and upload a new version through your Author Centre.

Kind regards,
Dr Laura Smith
Publishing Editor, Journals

RSC Associate Editor:
Comments to the Author:

(There are no comments.)

RSC Subject Editor:
Comments to the Author:
(There are no comments.)

Reviewer comments to Author:
Reviewer: 2

Comments to the Author(s)
Graphical Abstract needs to be redrawn

Author's Response to Decision Letter for (RSOS-192235.R1)

See Appendix B.

Decision letter (RSOS-192235.R2)

Dear Dr Bao:

Title: High-efficient fabrication of core-shell-shell structured $\text{SiO}_2@\text{GdPO}_4:\text{Tb}@\text{SiO}_2$ nanoparticles with improved luminescence
Manuscript ID: RSOS-192235.R2

It is a pleasure to accept your manuscript in its current form for publication in Royal Society Open Science. The chemistry content of Royal Society Open Science is published in collaboration with the Royal Society of Chemistry.

RSC Associate Editor
Comments to the Author:
The manuscript can now be accepted.

Reviewer(s)' Comments to Author:

Appendix A

Dear Editor,

Manuscript ID: RSOS-192235

Title: High-efficient fabrication of core-shell-shell structured SiO₂@GdPO₄:Tb@SiO₂ nanoparticles with improved luminescence

We sincerely thank the editor and all reviewers for their valuable comments. The manuscript has been explicitly revised according to the comments. The revisions have been highlighted with yellow in the manuscript. We have responded all the comments of reviewers by point to point, and the responses are listed as follows.

I hope that the revised manuscript will be acceptable for publication in the *Royal Society Open Science*. Please let me know of your decision at your earliest convenience.

Thank you for your time.

Yours sincerely,

JinrongBao

School of Chemistry and Chemical Engineering, University of Inner Mongolia, 235
Daxue Road, Saihan, Hohhot 010021, China

Tel.& Fax.: +86-0471-4992981

Email: jinrongbao@imu.edu.cn

First of all, we would like to express our appreciation for the reviewers' suggestive

comments. The following are our responses to the comments.

Responses to Reviewer 1:

Comments:

The paper content is confusing and has many questions. The paper may not be accepted.

Comment 1: In Fig. 1, the XRD patterns of $\text{SiO}_2@\text{GdPO}_4:\text{Tb}@\text{SiO}_2$ is difficult to confirm the successful synthesis of samples because they do not match.

Response: We thank the reviewer very much for his/her valuable comments.

In our work, the surface of SiO_2 is coated with a thin layer of $\text{GdPO}_4:\text{Tb}$ to save rare earth energy. The obtained $\text{SiO}_2@\text{GdPO}_4:\text{Tb}@\text{SiO}_2$ nanoparticles are consisted of SiO_2 core with a diameter of ~ 210 nm, $\text{GdPO}_4:\text{Tb}$ intermediate shell with thickness of ~ 7 nm, and SiO_2 outer shell with thickness of ~ 20 nm. Therefore, in the XRD patterns, the intensity of the diffraction peaks of SiO_2 is stronger, and the intensity of the diffraction peaks of $\text{GdPO}_4:\text{Tb}$ becomes weaker or even obscured. As shown in Figure R1 (Fig. 1 in the main manuscript), several weak XRD peaks at 25.90° , 26.90° , 28.68° and 30.94° were detected in the sample of $\text{SiO}_2@\text{GdPO}_4:\text{Tb}@\text{SiO}_2$, which were matched with monoclinic phase of GdPO_4 (JCPDS No.32-386) suggesting the successful synthesis of $\text{SiO}_2@\text{GdPO}_4:\text{Tb}@\text{SiO}_2$ nanoparticles.

Figure R1. XRD patterns of (a) SiO_2 , (b) $\text{SiO}_2@\text{GdPO}_4:\text{Tb}@\text{SiO}_2$.

Changes made to the manuscript:

1) The weak XRD peaks of GdPO_4 detected in the sample of $\text{SiO}_2@\text{GdPO}_4:\text{Tb}@\text{SiO}_2$ were marked in Fig. 1.

2) The following modification has been made in the revised manuscript (Page 2, lines 54-56).

“Fig. 1 showed the XRD patterns of SiO_2 and $\text{SiO}_2@\text{GdPO}_4:\text{Tb}@\text{SiO}_2$ nanoparticles. It can be seen that two diffraction peaks at $2\theta=8^\circ$ and 22° from amorphous SiO_2 were detected on both samples. Several new weak diffraction peaks appeared in $\text{SiO}_2@\text{GdPO}_4:\text{Tb}@\text{SiO}_2$, which were matched with monoclinic phase of GdPO_4 (JCPDS No.32-386).”

Comment 2: Excitation spectra showed that the strongest excitation peak of $\text{GdPO}_4:\text{Tb}$ nanoparticles appeared at 273 nm, while the core-shell-shell structured $\text{SiO}_2@\text{GdPO}_4:\text{Tb}@\text{SiO}_2$ appeared at 301 nm. Why?

Response: We thank the reviewer very much for his/her valuable comments.

We have retested the excitation spectrum of $\text{SiO}_2@\text{GdPO}_4:\text{Tb}@\text{SiO}_2$. As shown in

Figure R2, the retested plot is not consistent with previous one verifying that the strongest excitation peak was at 273 nm.

Figure R2. Previous test and retested excitation spectra of $\text{SiO}_2@\text{GdPO}_4:\text{Tb}@\text{SiO}_2$.

Changes made to the manuscript:

1) The previous excitation spectrum in Fig. S3 was replaced by the retested excitation spectrum of $\text{SiO}_2@\text{GdPO}_4:\text{Tb}@\text{SiO}_2$ in revised manuscript.

Comment 3: Excitation spectra should be shown in Fig. 5.

Response: We thank the reviewer very much for his/her valuable comments.

As shown in Figure R3, the excitation spectra were plotted together with emission spectra (Fig. 5 of the revised manuscript).

Figure R3. Excitation spectra (a) and emission spectra (b) of GdPO₄:Tb (blue line) and SiO₂@GdPO₄:Tb@SiO₂ (red line).

Comment 4: In Fig. S3, the excitation spectrum of SiO₂@GdPO₄:Tb@SiO₂ is questionable. It should be retested.

Response: We thank the reviewer very much for his/her valuable comments.

We have retested the excitation spectrum of SiO₂@GdPO₄:Tb@SiO₂. As shown in Figure R2, the retested plot is not consistent with previous one verifying that the strongest excitation peak was at 273 nm. The retested excitation spectrum of SiO₂@GdPO₄:Tb@SiO₂ was used to replace previous one in the revised manuscript.

Comment 5: In Fig. 5, the excitation wavelength of GdPO₄:Tb and SiO₂@GdPO₄:Tb@SiO₂ should all be 301 nm for their comparison. Why is the excitation wavelength 280 nm in Fig. 6?

Response: We thank the reviewer very much for his/her valuable comments.

The strongest excitation wavelength of BSA appeared at 280 nm [1]. Fig. 6 was the emission spectra of BSA with increase of the SiO₂@GdPO₄:Tb@SiO₂ nanoparticles concentration at different temperature at excitation wavelength of 280 nm. It has showed that the quenching of the tryptophan residues of BSA was done by the interaction between the as-prepared SiO₂@GdPO₄:Tb@SiO₂ nanoparticles and the BSA.

[1] Inci D, Koseler A, Zeytünlüoğlu A, Aydın R, Zorlu Y. Interaction of a new copper(II) complex by bovine serum albumin and dipeptidyl peptidase-IV. *J. Mol. Struct.* 2019, 1177 317-322.

Comment 6: What are the monitoring and excitation wavelength in Fig. S4?

Response: We thank the reviewer very much for his/her valuable comments.

The monitoring and excitation wavelength of GdPO₄:Tb nanoparticles were 543 nm and 273 nm. And the monitoring and excitation wavelength of SiO₂@GdPO₄:Tb@SiO₂ nanoparticles were 543 nm and 301 nm.

Responses to Reviewer 2:

Comments:

This manuscript reports the preparation of SiO₂@GdPO₄:Tb@SiO₂ core-shell-shell structure. The luminescence properties of this unique core-shell-shell structured were studied. The manuscript can be accepted for publication after major revision. But the following contents should be addressed.

Comment 1: The authors are encouraged to give the high-magnification and low-magnification FESEM images of samples.

Response: We thank the reviewer very much for his/her valuable comments.

We took high-magnification and low-magnification FESEM images of the samples, which were shown in Figure R4.

Figure R4. FE-SEM images of the products: (a,b) SiO₂, (c,d) SiO₂@GdPO₄:Tb, (e,f) SiO₂@GdPO₄:Tb@SiO₂ nanoparticles.

Changes made to the manuscript:

- 1) Figure R4 was added in the revised Supporting Information as Fig. S1.
- 2) The following description has been made in the revised manuscript (Page 2, line 65 and Page 3, Lines 1-4).

“The corresponding FESEM images of the as-synthesized products were shown in Fig. S1. It can be seen that SiO₂ spherical particles with an average size of 210 nm were non-aggregated and uniformly distributed (Fig. S1a and b). The diameter of SiO₂@GdPO₄:Tb increased to 225 nm after GdPO₄:Tb coating, and the surface become rougher (Fig. S1c and d). Furthermore, SiO₂@GdPO₄:Tb@SiO₂ still maintained a good spherical shape, while the particle size was increase to 265 nm (Fig. S1e and f).”

Comment 2: In Fig. 4, the schematic diagram of SiO₂@NH₂ is wrong.

Response: We thank the reviewer very much for his/her valuable comments.

We modified the schematic diagram of Fig. 4 in the revised manuscript as shown in Figure R5.

Figure R5. Schematic illustration showing the formation mechanism of core-shell-shell structured $\text{SiO}_2 @ \text{GdPO}_4:\text{Tb} @ \text{SiO}_2$.

Comment 3: What is the role of CTAB in the formation of SiO_2 shells?

Response: We thank the reviewer very much for his/her valuable comments.

In our work, SiO_2 outer shell was covered on the surface of $\text{SiO}_2 @ \text{GdPO}_4:\text{Tb}$ nanoparticles through the hydrolysis process of tetraethoxysilane (TEOS). After calcination, the core-shell-shell structured $\text{SiO}_2 @ \text{GdPO}_4:\text{Tb} @ \text{SiO}_2$ nanoparticles were obtained. Cetyl trimethyl ammonium bromide (CTAB) plays the role of the template agent in the formation of SiO_2 shells. When TEOS hydrolyzed to form the outer shell, CTAB formed a molecular layer on the surface of the silicon core in the reaction system, which is in favor of uniform hydrolysis and growth for TEOS. Meanwhile, because of the presence of CTAB, the SiO_2 layer produced a porous structure to prevent the structure from collapsing after calcination.

Changes made to the manuscript:

The following description has been added in the revised manuscript. (Page 3, lines: 20-22)

“Finally, SiO₂ outer shell was covered on the surface of SiO₂@GdPO₄:Tb nanoparticles in the presence of CTAB through the hydrolysis process of TEOS. CTAB formed a molecular layer on the surface of the silicon core in the reaction system, which would guarantee uniform hydrolysis and growth for TEOS.”

Comment 4: Experimental data of nitrogen adsorption-desorption should be provided.

Response: We thank the reviewer very much for his/her valuable comments.

The experimental data of nitrogen adsorption-desorption were plotted in the Figure R6. The BET surface are calculated from adsorption plot are 62 m²/g.

Figure R6. Nitrogen adsorption-desorption plot of SiO₂@GdPO₄:Tb@SiO₂.

Changes made to the manuscript:

- (1) Figure R6 has been added in the revised supporting information as Fig. S2.
- (2) The following description has been added in the revised manuscript (Page 3, Lines: 3-4).

“Meanwhile, SiO₂@GdPO₄:Tb@SiO₂ nanoparticles also had a high BET surface area of 62 m²/g.”

Comment 5: Fig. S2 is not clear and needs to be redrawn.

Response: We thank the reviewer very much for his/her valuable comments.

The resolution of Fig. S2 is enhanced. In our revised manuscript, Fig. S2 was rearranged as Fig. S4.

Comment 6: The authors should discuss in detail the effect of SiO₂ shell thickness on the luminescence performance of samples, and give the experimental results.

Response: We thank the reviewer very much for his/her valuable comments.

In our recent research on the core-shell-shell SiO₂@GdPO₄:Tb@SiO₂ nanoparticles, the effect of SiO₂ shell thickness on the luminescence performance has been studied. As shown in Figure R7, the emission intensity of the samples first increased with the increase of SiO₂ shell thickness, and then decreased. The maximum emission intensity located at the point where the SiO₂ shell thickness is about 20 nm. Therefore, we deposited SiO₂ shell with thickness of 20 nm to obtain an optimum synergistic effect between SiO₂ shell and GdPO₄:Tb layer.

Figure R7. The emission spectra of SiO₂@GdPO₄:Tb@SiO₂ nanoparticles with different thickness of SiO₂ shell, (a) ~20 nm, (b) ~30 nm and TEM image of the nanoparticles with outer shell thickness about 30 nm.

Responses to Reviewer 3:

Comments:

In this manuscript the authors reported the investigation of $\text{SiO}_2@\text{GdPO}_4:\text{Tb}@\text{SiO}_2$ nanoparticles with core-shell-shell structure by a silane coupling agent grafting method. Some interesting results are obtained. The core-shell-shell structured nanoparticles can effectively quench the intrinsic fluorescence of BSA through a static quenching mode. I therefore recommend an acceptance for publishing after next revisions.

Comment 1: Pages 2, Summary part, some background sentences can be simplified.

Response: We thank the reviewer very much for his/her valuable comments.

We have simplified some background sentences in revised the manuscript according to the reviewer's suggestion.

Comment 2: Introduction part, if possible, some important and relative reports about self-assembled core-shell-shell nanostructures (Composites Part B-Engineering, 2019, 164: 324-332; Applied Surface Science, 2020, 509: 145383.; Nanomaterials 2020, 10(1): 1.; Nanotechnology, 2020, 31(8): 085603.; Journal of Molecular Liquids, 2020, 298: 112010.) should be added to show clear background.

Response: We thank the reviewer very much for his/her valuable comments.

We have carefully read the above works. We found that the references of "Applied Surface Science, 2020, 509: 145383." And "Nanotechnology, 2020, 31(8): 085603." are very useful for our work in modifying the surface of the nanomaterials. Therefore, we cited these two references in our revised manuscript.

Comment 3: GA image should be added.

Response: We thank the reviewer very much for his/her valuable comments.

GA image (Figure R8) has been added in the revised manuscript.

Figure R8. GA image.

Changes made to the manuscript:

1) The following descriptions have been added in the revised manuscript (page 5).

“The core-shell-shell structure of $\text{SiO}_2@\text{GdPO}_4:\text{Tb}@\text{SiO}_2$ nanoparticles with effectively quenching the intrinsic fluorescence of bovine serum albumin were successfully synthesized”

Comment 4: What about the stability and reuse for this composite materials, please add more describe?

Response: We thank the reviewer very much for his/her valuable comments.

The SiO_2 shell can protect the phosphate materials from the surrounding environment. In consequence, the stability of $\text{SiO}_2@\text{GdPO}_4:\text{Tb}@\text{SiO}_2$ nanoparticles was increased. Furthermore, $\text{SiO}_2@\text{GdPO}_4:\text{Tb}@\text{SiO}_2$ nanoparticles can be reused after calcination. In other words, the adsorbed proteins and biomolecules can be removed from the surface of nanoparticles after heat treatment the functionalized $\text{SiO}_2@\text{GdPO}_4:\text{Tb}@\text{SiO}_2$ nanoparticles.

Changes made to the manuscript:

1) The following description has been added in the revised manuscript (Page 2, Lines :

4-6 and Page3, Line : 26-28).

“In addition, silica shell can greatly improve the stability of $\text{SiO}_2@\text{GdPO}_4:\text{Tb}@\text{SiO}_2$ nanoparticles through protecting the core materials from dissolution or hydrolysis.”

“Furthermore, functionalized $\text{SiO}_2@\text{GdPO}_4:\text{Tb}@\text{SiO}_2$ nanoparticles can be reused after calcination. In other words, the adsorbed proteins and biomolecules can be removed from the surface of nanoparticles after heat treatment the functionalized $\text{SiO}_2@\text{GdPO}_4:\text{Tb}@\text{SiO}_2$ nanoparticles.”

Comment 5: Some minor Language error should be modified;

Response: We thank the reviewer very much for his/her valuable comments.

The paper has been thoroughly checked and revised; the grammatical and syntactical errors have been revised.

Appendix B

Dear Editor,

Manuscript ID: RSOS-192235

Title: High-efficient fabrication of core-shell-shell structured SiO₂@GdPO₄:Tb@SiO₂ nanoparticles with improved luminescence

We sincerely thank the editor and all reviewers for their valuable comments. The manuscript has been explicitly revised according to the comments. We have responded the comments of reviewers, and the responses are listed as follows.

I hope that the revised manuscript will be acceptable for publication in the *Royal Society Open Science*.

Thank you for your time.

Yours sincerely,

Jinrong Bao

School of Chemistry and Chemical Engineering, University of Inner Mongolia, 235 Daxue Road, Saihan, Hohhot 010021, China

Tel.& Fax.: +86-0471-4992981

Email: jinrongbao@imu.edu.cn

Responses to Reviewer 2:

Comment 1: Graphical Abstract needs to be redrawn.

Response: We thank the reviewer very much for his/her valuable comment. The Graphical Abstract image was redrawn in our revised manuscript. As shown in Figure R1.

Figure R1 Graphical Abstract image